# OpenReview forum: "The Truthfulness Spectrum Hypothesis"
_ICML.cc/2026/Conference — Submitted to ICML 2026_

### Official Review · Reviewer_NKJL · 2026-02-15

**Soundness:** 2
**Presentation:** 1
**Significance:** 3
**Originality:** 3
**Overall Recommendation:** 4
**Confidence:** 3

**Summary:**

This paper proposes the Truthfulness Spectrum Hypothesis, arguing that large language models do not encode “truthfulness” as either a single universal direction or as entirely unrelated, task-specific signals; instead, truthfulness is represented along a spectrum ranging from highly general to highly domain- and context-specific components. Using a controlled dataset spanning multiple “truth types” (definitional, empirical, logical, fictional, ethical) plus challenging deceptive settings like sycophantic lying and expectation-inverted prompts, the authors show that linear “truth probes” often generalize well across the basic truth types but fail dramatically on sycophancy/expectation inversion, consistent with those directions being nearly orthogonal in representation space.

**Compliance With Llm Reviewing Policy:**

Affirmed.

**Final Justification:**

My concerns are removed, but I am not the expert of this area. I do not hold strong confident for my reviewer result.

**Key Questions For Authors:**

No

**Limitations:**

yes

**Strengths And Weaknesses:**

The topic is interesting and important. However, I have some severe issues for this paper:

**(1) Presentation**

the paper is very difficult to read, and the overall logic is hard to follow. Several sentences are overly academic and unnecessarily complex.

Key concepts are also poorly introduced and organized throughout the text. For instance, the term "geometric analysis" first appears on page 2 (line 96) but this phrase does not resurface until page 17, leaving a significant gap that disrupts the reader's understanding. Similarly, "sycophancy"—one of the most central concepts in the paper—lacks a clear and accessible definition where it is first introduced. For a term so fundamental to the paper's argument, readers should not have to search extensively for its meaning.

**(2) Truthfulness Spectrum Hypothesis**

The idea of probing truthfulness representations is both interesting and important. Based on their findings, the authors propose the "Truthfulness Spectrum Hypothesis," claiming that *"LLMs encode truthfulness along a spectrum of generality, with directions at varying levels of generality coexisting in the representational space."*

However, the results in Figure 5 closely resemble a standard pattern commonly seen in Singular Value Decomposition (SVD): the first principal direction captures the dominant shared signal across mixed data, while the remaining components capture heterogeneous, dataset-specific variation. If the observed "spectrum" is merely an artifact of this well-known decomposition behavior rather than a genuine property of truthfulness encoding, the paper's central claim would be substantially undermined. The authors should provide stronger evidence or additional controls to rule out this alternative explanation.

---

> ### Author Rebuttal · Authors · 2026-03-31
>
> We thank the reviewer for recognizing the importance of our research topic. We appreciate the feedback on presentation and the insightful question about the SVD interpretation, which we address in detail below.
>
> **W1: Presentation clarity**
> >"The paper is very difficult to read, and the overall logic is hard to follow. Key concepts are poorly introduced."
>
> We thank the reviewer for this feedback and have worked to improve the presentation. Specifically, we will add a clearer definition of sycophancy in the introduction and Section 2.2 and include a figure illustrating the sycophancy dataset construction process. We will also improve the flow between the geometric analysis preview in the introduction and the detailed treatment in Sections 5 and 6. We will also simplified several complex sentences throughout the paper.
>
> **W2: SVD alternative explanation for the spectrum**
> >"The results in Figure 5 closely resemble a standard pattern commonly seen in SVD: the first principal direction captures the dominant shared signal, while remaining components capture dataset-specific variation. If the observed 'spectrum' is merely an artifact of this decomposition behavior, the paper's central claim would be substantially undermined."
>
> We appreciate the reviewer for raising this question and the opportunity to clarify. The SVD/PCA explanation would predict that truth directions align with the high-variance principal components of the data. However, this is not what we observe.
>
> **[NEW] Truth directions are extremely low-variance directions.** The domain-general direction only explains 0.04% variance in the data. In addition, probes trained on individual domains explain *more variance* than the domain-general probe (0.48% for empirical truth direction vs. 0.04% for domain-general direction), the opposite of the proposed SVD explanation. Therefore, the truth spectrum reflects genuinely interesting structure in how LLMs encode different types of truthfulness, not a trivial decomposition artifact.
>
> New figure: variance explained for top PCs vs. probe directions: https://anonymous.4open.science/r/icml-rebuttal-25E1/var_explained_all.pdf
>
> Furthermore, Stratified INLP uses an iterative null-space projection procedure that explicitly differs from SVD: it finds directions that maximize classification accuracy across domains, not directions of maximum variance. The resulting domain-general directions are optimized for cross-domain truthfulness detection, a fundamentally different objective from principal component extraction.
>
> **The phenomenon may be more general, which strengthens our contribution.** The reviewer raises the possibility that the spectrum structure could arise as a general property of linear encoding rather than being specific to truthfulness. We find this an interesting point for future work: if this is a general principle of how LLMs linearly encode multi-domain concepts, that would make the finding more significant, not less. To our knowledge, no existing work has demonstrated or characterized such a spectrum structure with respect to linear classification in LLM representations. We focus on truthfulness because of its direct connection to alignment and AI safety, but extending this analysis to other concepts (e.g., sentiment, toxicity, intent) is a promising direction (see Discussion L399-403).
>
> **Additional novel results:**
>
> We significantly strengthen Section 5 with the Mahalanobis cosine similarity metric, which achieves R2 = 0.98 in predicting cross-domain generalization, far exceeding standard cosine similarity (R2 = 0.56). We further validate this relationship with controlled simulations across five diverse data distributions (isotropic/anisotropic Gaussian, leptokurtic, leptokurtic+skewness). Mahalanobis cosine similarity achieves R2 > 0.95 in all conditions, while standard cosine similarity’s R2 drops to as low as 0.01.
>
> New figure: https://anonymous.4open.science/r/icml-rebuttal-25E1/geo_ood_vs_mcossim.pdf

---

> > ### Author Rebuttal · Reviewer_NKJL · 2026-04-03
> >
> > Thank you very much for your responses. I appreciate the effort that the authors put into it. My concerns are removed.

---

### Official Review · Reviewer_4UAS · 2026-02-21

**Soundness:** 3
**Presentation:** 3
**Significance:** 3
**Originality:** 3
**Overall Recommendation:** 4
**Confidence:** 3

**Summary:**

This paper studies how truthfulness is represented inside LLMs and argues that truth directions form a **spectrum**: some are domain-general, others are domain-specific, and some are intermediate.
To test this, the authors build FLEED (five truth types: definitional, empirical, logical, fictional, ethical), add sycophancy and expectation-inverted lying datasets, and evaluate probe transfer across domains and existing honesty benchmarks.
Main findings: (1) probes transfer well across most FLEED domains but fail on sycophancy/expectation-inverted settings, (2) joint training across all domains recovers strong general performance, (3) probe geometry and concept-erasure analyses support mixed general/specific structure, and (4) post-training pushes sycophancy representations further away from other truth types. The paper also includes causal steering experiments suggesting domain-specific directions are more useful for intervention than domain-general ones.

**Compliance With Llm Reviewing Policy:**

Affirmed.

**Final Justification:**

My concerns have almost been solved.

**Key Questions For Authors:**

1. How do you rule out synthetic-generation artifacts in FLEED (style cues, templating, or source-model fingerprints) as a driver of transfer behavior?
   A stronger artifact-control analysis could raise my confidence.
2. Can you more cleanly separate “general direction exists” from “joint training simply improves coverage/data efficiency”?
   For example, controlled data-matching experiments would help.
3. How stable are the causal intervention results across models, layers, and intervention strengths?
   Broader robustness here could materially strengthen the intervention claims.
4. Do the same geometric conclusions hold on more natural, human-authored truthfulness datasets with minimal prompt engineering?

**Limitations:**

The paper includes a clear limitations section and discusses dual-use concerns. Minor additions on synthetic-data bias and intervention robustness would make it stronger.

**Strengths And Weaknesses:**

### Strengths
1. The paper addresses an important and timely question in LLM honesty: why prior work finds both strong generalization and sharp failures.
2. The empirical scope is broad: multiple truth types, deception settings, external benchmarks, and base-vs-chat comparisons across model families.
3. The analysis goes beyond standard probing: direction similarity, Stratified INLP, LEACE-based selective erasure, and causal intervention are thoughtfully combined.
4. The conclusions are practically useful: broad detectors may exist, but robust behavioral steering appears more domain-dependent.

### Weaknesses
1. A large part of FLEED is model-generated; this raises concerns about synthetic artifacts and distributional shortcuts.
2. The claim that a domain-general truth direction “exists” relies heavily on joint-training success; alternative explanations (data/coverage effects, superposition) are not fully ruled out.
3. Causal intervention evidence is relatively narrow (mainly one model/setting) and effect sizes are modest.
4. The study focuses on linear structure; nonlinear representations may change the interpretation.
5. Stronger statistical reporting (e.g., uncertainty intervals and more systematic significance testing in key comparisons) would improve confidence.

---

> ### Author Rebuttal · Authors · 2026-03-31
>
> We thank the reviewer for the thoughtful assessment and for highlighting the importance and timeliness of our research question, the breadth of our empirical scope, and the practical utility of our conclusions for lie detection and alignment. We address each concern below with new experiments and clarifications.
>
> **W1: Ruling out synthetic-generation artifacts in FLEED**
> >"How do you rule out synthetic-generation artifacts in FLEED (style cues, templating, or source-model fingerprints) as a driver of transfer behavior?"
>
> **[NEW]** We include results using Claude as the generation model (in addition to Gemini), showing that the cross-domain generalization patterns are consistent regardless of the generation source. If probes were exploiting model-specific fingerprints, we would expect substantially different transfer patterns across generation models.
>
> New figure: https://anonymous.4open.science/r/icml-rebuttal-25E1/robustness_gen_claude_fled.pdf
>
> **[NEW]** We show cross-domain generalization performance and post-training effects with prior independently constructed datasets. First, we show that probe generalize to held-out domains of Marks & Tegmark, 2023; Azaria & Mitchell, 2023. We train on FLEED + Sycophancy + Goldowsky-Dill et al., 2025 and test on on-policy datasets (roleplaying, insider trading, sandbagging) and prior factual truthfulness datasets (Marks & Tegmark, 2023; Azaria & Mitchell, 2023). These datasets were constructed with different methodologies, templates, and models, yet probes transfer well between them and FLEED. This suggests that the probe picks up on genuine truth features beyond superficial artifacts like “style cues, templating, or source-model fingerprints.”
>
> New figure: https://anonymous.4open.science/r/icml-rebuttal-25E1/robustness_gen_unseen_domains.pdf
>
> >"Do the same geometric conclusions hold on more natural, human-authored truthfulness datasets with minimal prompt engineering?"
>
> We also show the same post-training reorganization effects using popular datasets from prior works (Marks & Tegmark, 2023; Azaria & Mitchell, 2023) instead of FLEED. This largely rules out the possibility that transfer is driven by shared generation artifacts.
>
> New figure: https://anonymous.4open.science/r/icml-rebuttal-25E1/robustness_posttraining_data.pdf
>
> Additionally, our dataset construction uses diverse negation strategies (direct negation, term replacement, quantity modification, information substitution) specifically to reduce the risk of probes learning surface-level patterns. We also manually filter for quality.
>
> **W2: Separating “general direction exists” from “joint training improves coverage”**
> > "Can you more cleanly separate ‘general direction exists’ from ‘joint training simply improves coverage/data efficiency’?"
>
> **[NEW]** To control for data efficiency, we evaluate the domain-general probe trained on the full training set versus only 30% of the training data (~4k samples, comparable in size to individual datasets such as FLED or sycophancy). As shown below, performance remains high across nearly all domains, with only modest degradation in sycophancy. This is consistent with the well-known property that linear probes converge quickly and are sample-efficient.
>
> | Training Data | Definitional | Evidential | Logical | Fictional | Ethics | Honesty | Roleplaying | Insider Trading | Sandbagging | Sycophancy |
> |:--|:--:|:--:|:--:|:--:|:--:|:--:|:--:|:--:|:--:|:--:|
> | 100% | 0.997 | 0.999 | 0.956 | 0.993 | 0.989 | 0.946 | 0.967 | 0.999 | 1.000 | 0.953 |
> | 30%  | 0.996 | 0.999 | 0.934 | 0.992 | 0.983 | 0.929 | 0.921 | 0.999 | 1.000 | 0.870 |
>
> **Additional Results:**
>
> We significantly strengthen Section 5 with the Mahalanobis cosine similarity metric, which achieves R2 = 0.98 in predicting cross-domain generalization, far exceeding standard cosine similarity (R2 = 0.56). We further validate this relationship with controlled simulations across five diverse data distributions (isotropic/anisotropic Gaussian, leptokurtic, leptokurtic+skewness). Mahalanobis cosine similarity achieves R2 > 0.95 in all conditions, while standard cosine similarity’s R2 drops to as low as 0.01.
>
> New figure: https://anonymous.4open.science/r/icml-rebuttal-25E1/geo_ood_vs_mcossim.pdf

---

> > ### Author Rebuttal · Reviewer_4UAS · 2026-04-06
> >
> > Thanks for the response. My concerns have almost been solved.

---

### Official Review · Reviewer_MWZ9 · 2026-03-12

**Soundness:** 2
**Presentation:** 3
**Significance:** 3
**Originality:** 2
**Overall Recommendation:** 3
**Confidence:** 4

**Summary:**

This paper proposes the *Truthfulness Spectrum Hypothesis*, which states that the representational space in language models contains directions ranging from broadly domain-general to narrowly domain-specific. It introduces the FLEED dataset with true and false statements from five different domains. Using this, and other datasets, it investigates the geometric relationship and predictive power of different directions related to truthfulness. Directions are identified through multiple experiments involving the training of linear probes on subsets of FLEED and additional datasets, alongside methods like LEACE, and the newly proposed Stratified INLP, which isolates both specific and general truthfulness directions. Further analysis investigates how sycophancy representations differ between base and post-trained models.

**Compliance With Llm Reviewing Policy:**

Affirmed.

**Final Justification:**

The authors did address some of my concerns during the rebuttal, but the remaining ones I list in my last comment are significant enough that my final assessment remains unchanged.

**Key Questions For Authors:**

1. Is there evidence that the domain-specific directions are not merely dataset-specific confounders, that is, do they generalize between differently constructed datasets of the same domain, for example, datasets created by prior work?
2. In cases where pairwise transfer of probes between domains fails, is there evidence that the jointly trained probes do not merely track linear combinations of domain-specific directions, rather than truly domain-general directions in the sense that they generalize to unseen distributions? Relatedly, do directions of intermediate generality identified by LEACE carry semantic meaning, or could they also be explained as linear combinations of general and domain-specific directions?
3. Given that causal interventions along the directions identified by Stratified INLP have mixed effects on the model's discriminative power of true and false statements (Section 8), how does Stratified INLP compare to alternative methods for separating general from specific directions? Specifically, can the results be explained by the general directions identified by Stratified INLP being linear combinations with a high positive coefficient for a truly domain-general direction and smaller positive coefficients for truly domain-specific directions; and consequently, the specific directions being linear combinations with large positive coefficients for truly domain-specific directions and, due to the orthogonality constraint, small negative coefficients for the truly domain-general direction? Additionally, do you add or subtract the scaled truth direction (α = −2.0), and which parts of the results speak for and against interpreting the identified directions as truth directions?

**Limitations:**

Yes. Limitations are discussed in Section 11, with valid points addressed adequately. Additional limitations are mentioned throughout the paper.

**Strengths And Weaknesses:**

## Soundness

### Strengths

- The paper features a wide range of experiments, demonstrating awareness of key datasets and methods used in the field.
- The paper substantiates its findings using four models plus base models.

### Weaknesses

- Only one dataset is used for each domain, which leaves doubt whether “domain-specific” directions discussed throughout the paper merely capture superficial, “dataset-specific” phenomena. For example, the domain-specific direction found by Stratified INLP on the sycophancy dataset could merely represent “Repetition of an option that has been mentioned in the prompt”, instead of a sycophancy-specific direction that generalizes across diverse types of sycophancy.
- A core proposition of the paper is that the existence of a general truthfulness direction can not be ruled out. However, the evidence provided by this paper to reject arguments made by existing work is insufficient.
  - Example: “Prior work has treated cross-domain generalization failure and geometric dissimilarity between probes as evidence against domain-general truth encoding. However, this inference is flawed: generalization may fail simply because more diverse data is needed to discover the domain general directions.” This criticism seems misplaced, as a truly general truth direction should, by definition, be present even in narrow data. If the authors merely mean that isolating this direction requires diverse data, it does not seem to contradict the claims by prior work and should be clarified.
  - Example: “Yet training on all domains jointly recovers strong performance, confirming that domain-general directions exist despite poor pairwise transfer”. While this shows that a direction correlating with truthfulness across the training datasets exists, the result can be explained by the probe tracking a linear combination of domain-specific directions of each training dataset, rather than a domain-general direction.
  - More broadly, the truthfulness spectrum hypothesis as stated — that "the representational space contains directions ranging from broadly domain-general to narrowly domain-specific" — is trivially satisfiable whenever multiple near-orthogonal dataset-specific directions exist, since linear combinations of such directions will by construction exhibit varying degrees of cross-dataset performance. The hypothesis becomes meaningful only if these directions generalize to unseen domains, but the paper does not test this. The combined probe (Figure 2) is trained and evaluated on the same set of domains (via cross-validation), so its strong cross-domain performance cannot distinguish a genuinely domain-general direction from a linear combination of domain-specific ones. A straightforward test — training on a subset of domains and evaluating on a fully held-out domain — is missing.
The paper states: “For interventions, our causal experiments show that domain-specific directions outperform domain-general ones, suggesting that while domain-general probes enable broad detection, they may be limited for reliable behavioral control.” This seems to underappreciate the fact that these interventions had opposite effects of comparable magnitude, which remains underinvestigated.

## Presentation

### Strengths

- The methods are clearly described enabling reproducibility.
- Design decisions like which layer to probe on are clearly justified and supported by data in the appendix.
- The paper is well-written with comprehensive figures and tables.
- The paper provides a helpful example for each subset of the FLEED dataset.
- The paper is well-contextualized with prior work.

### Weaknesses

- The following section is confusing: “We apply stratified INLP on our FLEED and sycophancy datasets and factual knowledge datasets from prior works (Azaria & Mitchell, 2023; Marks & Tegmark, 2023; Goldowsky-Dill et al., 2025) to obtain 14 domain-specific directions and a single general direction. We run with 10 general directions and 5 domain-specific ones.” It is unclear how exactly the directions are determined, and how do the directions in these sentences relate to each other. In particular, there seems to be a mismatch: if only a single general direction was identified, which 10 general directions are being run with?
- In section 8, the paper states: “To intervene in the model behavior, we add a scaled truth direction d to the MLP output bias at layer 15: b′ℓ = bℓ +α · d, with α = −2.0.” According to a negative alpha, this looks like the truth direction is subtracted rather than added. It is unclear what the expectations in the following analysis are, in particular, whether a positive delta is evidence for or against d being similar to the model’s representation of truthfulness.

## Significance

### Strengths

- The investigation of representations of truthfulness in language models is a relevant topic with implications for the safety of AI systems that deserves attention.
- The FLEED dataset is a meaningful contribution with novel properties to enable investigation of how different types of truthfulness are represented in language models.
- The insight that probes generalize across subsets of FLEED extends work of probes generalizing across domains
- The investigation of how post-training shapes representations of truthfulness is meaningful and may be followed-up by similar work targeting domains beyond sycophancy.

### Weaknesses

- I am concerned that the main contribution of the paper is merely showing that one can linearly combine directions that predict truthfulness on specific datasets with directions that generalize across datasets to obtain directions that selectively predict truthfulness on a subset of these datasets. To be clear: the pairwise transfer results (e.g., definitional → empirical) in Figure 2 do demonstrate genuine cross-domain generalization for some domains, but this has been shown for other domains in prior work (e.g., Marks & Tegmark, 2023). The novel claims about the spectrum — particularly the domain-general and intermediate-generality directions — lack evidence of generalization to unseen domains, which is where the contribution would need to go beyond prior findings.

## Originality

### Strengths

- Propose a new method Stratified INLP to identify general and specific directions across datasets.
- Novel analysis of representations of sycophancy in base vs. chat models.
- The novel FLEED dataset introduces true and false statements across intuitively distinct types of truthfulness for which datasets were lacking.

### Weaknesses

- Some key findings are extensions of prior work, for example, that some directions in a language model’s representations generalize across some domains.

---

> ### Author Rebuttal · Authors · 2026-03-31
>
> We thank the reviewer for their detailed engagement!
>
> **W1: Domain-specific vs. dataset-specific directions**
> >"Only one dataset is used for each domain, which leaves doubt whether ‘domain-specific’ directions merely capture superficial, ‘dataset-specific’ phenomena."
>
> Our causal experiments directly address this: domain-specific directions extracted by Stratified INLP yield a positive mean causal effect on SimpleQA (a held-out dataset with a different format), showing they encode genuinely truth-relevant features rather than spurious ones.
>
> >"The hypothesis becomes meaningful only if these directions generalize to unseen domains, but the paper does not test this."
>
> **[NEW]** We include experiments training on some datasets (left) and testing on fully held-out on-policy domains (insider trading, sandbagging, roleplaying) and prior factual truthfulness benchmarks (right). This directly shows strong generalization to unseen domains.
>
> New figure: https://anonymous.4open.science/r/icml-rebuttal-25E1/robustness_gen_unseen_domains.pdf
>
> **W2: Linear combination of domain-specific directions?**
> >"The result can be explained by the probe tracking a linear combination of domain-specific directions of each training dataset, rather than a domain-general direction."
>
> We address this at three levels: First, we define domain-general for detection/monitoring purposes. If a direction detects lies across all domains, it is domain-general for lie detection. Second, generalization to unseen domains (W1 above) would not be expected from a mere linear combination of training-domain directions. Third, combining orthogonal domain-specific directions does not trivially yield a full spectrum. It critically depends on nuisance variable distributions within each domain.
>
> **W3: Prior work criticism is misplaced**
> >"A truly general truth direction should, by definition, be present even in narrow data. If the authors merely mean that isolating this direction requires diverse data, it does not seem to contradict the claims by prior work."
>
> Multiple prior works explicitly interpret narrow-data probe failures as evidence that no domain-general truth direction exists. For example:
>
> > Orgad et al. (2024): “probing classifiers do not generalize across different tasks...challenging the idea of a "universal truthfulness" encoding...Instead, our results indicate that LLMs encode multiple, distinct notions of truth.”
>
> > Azizian et al. (2025): “We observe that these ‘geometries of truth’ are intrinsically task-dependent and fail to transfer across tasks...LLMs likely have multiple geometries of truth, but that they are irreconcilable and highly task-dependent.”
>
> Our paper challenges this inference: a domain-general direction can exist yet not be recoverable from narrow data.
>
> **W4: Causal interventions have mixed effects**
>
> Clarification: the probe is trained to detect lies, so positive values correspond to lying. A negative alpha steers the model toward honesty. We will make it clear in revision.
>
> > “This seems to underappreciate the fact that these interventions had opposite effects of comparable magnitude, which remains underinvestigated.”
>
> The domain-general direction likely captures variance from both factual and sycophancy domains; steering with it on a purely factual test introduces irrelevant sycophancy variance and degrades performance, consistent with prior work (Turner et al., 2024). Importantly, the magnitudes are not comparable as suggested: the best-matched domain-specific directions produce substantially stronger positive effects than the domain-general direction's negative effect (+0.35 vs. −0.11 at top confidence), supporting our claim.
>
> **Contributions of this paper**
>
> We respectfully disagree that the main contribution is “merely showing that one can linearly combine directions.” Our main contributions are:
> 1) New carefully controlled datasets (FLEED, sycophancy)
> 2) Joint directions succeed even when cross-domain probes are orthogonal or even anti-correlated
> 3) The truthfulness spectrum hypothesis reconciling contradictory prior findings
> 4) Novel method: Stratified INLP
> 5) **[NEW]** Mahalanobis cosine similarity predicting generalization with R2=0.98
> 6) Post-training reorganizes truthfulness geometry
> 7) Steering experiments showing domain-specific directions are causally relevant and better than domain-general one
>
> **Additional Results:**
>
> We significantly strengthen Section 5 with the Mahalanobis cosine similarity between directions, which achieves R2=0.98 predicting OOD generalization, far exceeding standard cosine similarity (R2=0.56). We further validate this relationship with controlled simulations across five diverse data distributions (isotropic/anisotropic Gaussian, leptokurtic, leptokurtic+skewness). Mahalanobis cosine similarity achieves R2 > 0.95 in all conditions, while standard cosine similarity’s R2 drops to as low as 0.01.
>
> New figure: https://anonymous.4open.science/r/icml-rebuttal-25E1/geo_ood_vs_mcossim.pdf

---

> > ### Author Rebuttal · Reviewer_MWZ9 · 2026-04-03
> >
> > Thank you for your responses.
> >
> > Resolved doubts:
> > * The new transfer experiments do show that meaningful FLEED-general directions exist that are not just linear combinations.
> > * I agree that for detection/monitoring purposes, the question of whether a direction is meaningful or a linear combination is irrelevant.
> >
> > Remaining doubts:
> > * The “domain general” directions may just be FLEED-general \- datasets that all share similar structure and origin. As you have shown, probes trained on subsets of FLEED already transfer to other subsets. For a direction to be truly “domain-general” and meaningful, which your hypothesis seemingly suggests, it should transfer to unseen datasets that do not have this property (e.g., Azizian et al., 2025).
> > * My concern that “domain-specific” directions may not generalize across structurally different datasets of the same domain is still unaddressed \- whether some such domain-specific directions have some effect on SimpleQA is not sufficient to refute this.
> > * There is tension between the paper’s narrative, which seems to suggest that the domain-general directions are meaningful, and the response to W3, that for detection/monitoring purposes, meaningfulness does not matter.
> >
> > Given the above remaining concerns, I have maintained my score.

---

> > > ### Author Response · Authors · 2026-04-06
> > >
> > > We thank the reviewer for the thoughtful follow-up and for clarifying the remaining concerns. We address each point below.
> > >
> > > **Domain-general directions being merely FLEED-general**
> > >
> > > We believe our previous rebuttal experiment has already directly addressed this concern. Specifically, we trained on FLEED + Sycophancy + Goldowsky-Dill et al. (2025) datasets [left of figure] and tested on held-out, independently constructed datasets: on-policy domains (roleplaying, insider trading, and sandbagging) and prior truthfulness datasets (Marks & Tegmark, 2023; Azaria & Mitchell, 2023) [right of figure]. These test sets differ from train sets in both origin and structure (even generalizing from pre-filled off-policy datasets to on-policy datasets sampled from the model’s logits), yet the domain-general direction transfers successfully. This is exactly the kind of evidence the reviewer requested: domain-general probe generalizing to unseen datasets that do not share FLEED's structure or origin.
> > >
> > > [Figure from previous rebuttal](https://anonymous.4open.science/r/icml-rebuttal-25E1/robustness_gen_unseen_domains.pdf)
> > >
> > > **Whether domain-specific directions generalize across structurally different datasets of the same domain**
> > >
> > > **[NEW]** We show that domain-specific directions extracted from FLEED (Gemini-generated) using stratified INLP transfer to independently constructed datasets of the same domain but not to datasets of other domains. Specifically, the empirical-specific direction generalizes to Claude-generated empirical truth dataset and to empirical claims from Azaria & Mitchell (2023) — datasets differing in methodology, format, and topic coverage — but fails on logical datasets (logical from FLEED, Claude-generated logical dataset, and logical samples from Marks & Tegmark, 2023). Conversely, the logical-specific probe generalizes to the Claude-generated logical dataset and logical samples from Marks & Tegmark (2023), but not to empirical datasets. This selective transfer confirms that these directions capture domain-level structure rather than dataset-specific artifacts.
> > >
> > > [New Figure](https://anonymous.4open.science/r/icml-rebuttal-25E1/prob_gen_domain_specific.pdf)
> > >
> > > More broadly, perfect within-domain transfer is not a requirement of our argument. Stratified INLP isolates directions maximally predictive of one dataset after removing shared directions, so the resulting directions could be more specific than the intuitive domain categories we define (empirical, definitional, etc.). However, this does not contradict our claims. Our claim is that these directions occupy one end of the generality spectrum: they encode truth-relevant structure (confirmed by causal steering on held-out SimpleQA) while being far more narrowly tuned than domain-general directions. Whether a direction transfers across all datasets within a domain category simply determines how far toward the specificity end of the spectrum it falls.
> > >
> > > **Tension between "meaningfulness" and detection/monitoring**
> > >
> > > We respectfully note that our paper has been consistent on this point from the outset. We explicitly distinguish between directions that are *meaningful for detection/monitoring* and directions that are *causally meaningful* for steering behavior, and we focus on the former (Introduction, p. 2, L106–109). Our claim is that the domain-general directions we identify are meaningful in the sense that they reliably detect truthfulness across diverse domains, and our experiments thoroughly support this. That said, we appreciate the reviewer raising this point, and will make the distinction between *detection-meaningful* and *causally-meaningful* directions more prominent in the revision.
> > >
> > > We also acknowledge that understanding the exact causal role of *detection-meaningful* domain-general directions and the mechanism behind them being predictive yet ineffective for steering is an important open question, but one that falls outside the scope of this work and constitutes a promising direction for future research. We will expand upon this in our revised discussion section.

---

### Official Review · Reviewer_s2gm · 2026-03-13

**Soundness:** 4
**Presentation:** 3
**Significance:** 3
**Originality:** 3
**Overall Recommendation:** 4
**Confidence:** 2

**Summary:**

This paper studies how truthfulness is represented in the hidden states of large language models. The authors propose the Truthfulness Spectrum Hypothesis, suggesting that truth representations exist along a continuum ranging from domain-general directions to domain-specific ones in representation space. To test this hypothesis, the paper constructs the FLEED dataset containing multiple truth categories and with additional datasets targeting sycophantic and expectation-inverted lying scenarios. The authors train linear probes on model activations to analyze cross-domain generalization of truth representations. They further analyze probe geometry, study the effects of post-training, and introduce a stratified INLP method to isolate domain-general and domain-specific truth directions.

**Compliance With Llm Reviewing Policy:**

Affirmed.

**Final Justification:**

I appreciate the authors’ rebuttal and the effort they put into addressing my questions. After considering the paper and the rebuttal, I maintain my original evaluation and recommendation.

**Key Questions For Authors:**

The results suggest post-training reshapes the geometry of truth representations. Could the authors clarify which components of post-training contribute most to this effect?

**Limitations:**

yes

**Strengths And Weaknesses:**

Strengths:

1.  The analysis combining probe generalization, vector geometry, and concept-erasure techniques provides useful empirical insights into how truth-related signals may be structured in representation space.
2.  The proposed Stratified INLP procedure is an interesting methodological extension for isolating domain-general and domain-specific directions.

Weaknesses:

1. The spectrum hypothesis mainly provides a conceptual interpretation of the observed probe behaviors. It is unclear whether the experimental results uniquely support this hypothesis compared to simpler explanations such as dataset-dependent probe alignment.
2. The conclusions rely heavily on probe direction similarity and cross-domain transfer. It would be helpful to understand how robust these geometric findings are under alternative probing setups.
3. The interpretation of the causal steering results remains unclear, particularly why the domain-general direction degrades performance despite being predictive across domains.

---

> ### Author Rebuttal · Authors · 2026-03-31
>
> We thank the reviewer for recognizing the breadth of our analysis combining probe generalization, vector geometry, and concept-erasure techniques, and for highlighting Stratified INLP as an interesting methodological contribution. We address each concern below.
>
> **W1: Robustness under alternative probing**
> > "It would be helpful to understand how robust these geometric findings are under alternative probing setups."
>
> **[NEW] Robustness under different probing setups.** We replicate our main results (Figures 2 & 4) using Difference-of-Means and LDA probes. Core findings hold across all architectures: high cross-domain transfer within FLEED, failure on sycophancy/expectation-inverted datasets, and discovery of a domain-general direction. Post-training geometric reorganization is also robust. Some deviations also emerge (e.g., weak LDA transfer from ethical claims, poor DiffMean sycophancy performance, DiffMean is better for chat models than base models), which is expected since logistic regression was selected for best cross-domain performance during probe tuning (Appendix B).
>
> New figure: cross-domain generalization for LDA: https://anonymous.4open.science/r/icml-rebuttal-25E1/robustness_gen_lda.pdf
>
> New figure: cross-domain generalization for Diffmean: https://anonymous.4open.science/r/icml-rebuttal-25E1/robustness_gen_diffmean.pdf
>
> New figure: post-training results for LDA and Diffmean: https://anonymous.4open.science/r/icml-rebuttal-25E1/robustness_posttraining_probe.pdf
>
>
> **W2: Causal interpretation of domain-general vs. domain-specific directions**
> > "The interpretation of the causal steering results remains unclear, particularly why the domain-general direction degrades performance despite being predictive across domains."
>
> We explain this in Section 8 L354-358. We reiterate the two important points of interpreting the causal results:
> 1) **Domain-specific directions are causally relevant, not spurious.** Domain-specific directions improve factual discrimination on a fully held-out test set (SimpleQA) with a different QA format, ruling out dataset-specific confounders.
> 2) **Detection directions and steering directions can differ.** A direction that is linearly predictive of truthfulness across domains (useful for monitoring) need not be the same direction that causally controls the model’s truthful behavior (useful for steering). The domain-general direction captures both factual and sycophancy variance; the latter acts as noise when steering on factual tasks, explaining the degradation. Steering along a misaligned direction causing performance drops is consistent with prior work (Turner et al., 2024).
>
> **W3: Dataset-dependent probe alignment as a simpler explanation**
> > "The interpretation of the causal steering results remains unclear, particularly why the domain-general direction degrades performance despite being predictive across domains."
>
> We are unsure what it means. If the concern is that probes pick up on dataset-specific spurious features rather than genuine truth encoding, our causal experiments on SimpleQA (held-out, different format) address this directly: domain-specific directions encode genuinely truth-relevant features that transfer instead of mere confounds (Section 8).
>
> **[NEW]** We additionally show generalization to fully held-out domains by training on FLEED + Sycophancy + Goldowsky-Dill et al., 2025 datasets and testing on unseen on-policy domains (roleplaying, insider training, and sandbagging) and prior factual truthfulness datasets (Marks & Tegmark, 2023; Azaria & Mitchell, 2023). The probe shows strong generalization performance across all seen and unseen domains.
>
> New figure: generalization to unseen domains: https://anonymous.4open.science/r/icml-rebuttal-25E1/robustness_gen_unseen_domains.pdf
>
> **Q: Which components of post-training contribute most?**
>
> This is a very interesting question. But we do not have access to intermediate checkpoints to definitively isolate which component of post-training (instruction tuning, RLHF, etc.) drives the geometric reorganization. Investigating this with controlled post-training ablations is an important direction for future work.
>
> **Additional Results:**
>
> We significantly strengthen Section 5 with the Mahalanobis cosine similarity metric, which achieves R2 = 0.98 in predicting cross-domain generalization, far exceeding standard cosine similarity (R2 = 0.56). We further validate this relationship with controlled simulations across five diverse data distributions (isotropic/anisotropic Gaussian, leptokurtic, leptokurtic+skewness). Mahalanobis cosine similarity achieves R2 > 0.95 in all conditions, while standard cosine similarity’s R2 drops to as low as 0.01.
>
> New figure: Mahalanobis cosine similarity: https://anonymous.4open.science/r/icml-rebuttal-25E1/geo_ood_vs_mcossim.pdf

---

> > ### Author Rebuttal · Reviewer_s2gm · 2026-04-03
> >
> > Thank you very much for your responses. I appreciate the effort that the authors put into it.

---

### Decision · Program_Chairs · 2026-04-30

**Decision:**

Reject

**Comment:**

The authors investigate the representation geometry of language models, and study the similarities and differences in how factual information and sycophantic model behavior are represented. The key proposal is the truthfulness spectrum hypothesis, which attempts to add nuance to prior work on measuring "truth" in LLM internal activations through training linear probes. The authors provide fairly extensive empirical results, where linear probes are trained across several flavors of truthfulness datasets. Generalization, and failure to generalize, are analyzed in a variety of ways, including by designing probes that are domain-specific, as well as activation steering. Despite the authors putting a commendable amount of work during the rebuttal into broadening the set of empirical results, I share some of the initial concerns raised about the soundness of the probing and steering methodologies, and the extent to which this proposed framework moves us closer to actionable monitorability of capable nets. I am somewhat troubled by the fact that the hypothesis itself does not seem falsifiable by any method other than probing, and as Reviewer MWZ9 points out, the apparently rich structure contained across the various domain-specific probes could plausibly be simply due to the high dimensional nature of the representation itself.  The authors have included SVD results to rebut Reviewer NKJL's concern about a lack of alternative hypotheses considered. The results show that probe directions do not align with principal components. However in classical OLS regression or LDA classification, model tend to align their predictions precisely with such low variance directions: these are amplified due to the inversion of the data covariance matrix in the closed form solution, so the fact that probes do not "explain" the data in a PCA sense is not surprising.  In a very high-dimensional space, there will always be many low-variance directions, and the presence of any amount of noise will create spurious correlations with any binary label; very recent work critiquing probing comes to similar conclusion through a different methodology, by looking at probes on randomly initialized nets [http://arxiv.org/abs/2512.18792]. The authors show that RLHF amplifies the predictive power of probes, suggesting perhaps that models become more capable of strategic deception after alignment. However the use of sycophancy prompts to generate one evaluation dataset are set up so that "lying" is the same as adhering to user instructions, so I wonder if the representation geometry the authors discover tells us more about instruction following than it does about "truth", per se. While I appreciate the many empirical results offered by the authors (especially during the rebuttal), I ultimately feel they fall short of supporting the broad claim about how "truth" is organized internally within LLMs, and recommend against accepting the paper.